# Spatial Delimitation of Small Headwater Catchments and Their Classification in Terms of Runoff Risks

Petr Kavka 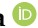

Department of Landscape Water Conservation, Faculty of Civil Engineering, Czech Technical University in Prague, Thákurova 7, Prague 6, 166 29 Prague, Czech Republic; petr.kavka@fsv.cvut.cz

**Abstract:** The hydrological similarity of catchments forms a basis for generalizing their hydrological response. This similarity of the hydrological response enables catchments to be classified from numerous perspectives, e.g., hydrological extremes or ecological aspects of catchments. A specific group is formed by so-called "first-order catchments". This article describes the derivation process of small headwater catchments up to 5 km$^2$ in size on the territory of the Czech Republic. The delimitation is based on the digital terrain model, the stream network, and the water reservoirs. The catchments derived in this way cover 80% of the country. Five mutually independent and sufficiently representative parameters were selected with Principal Components Analysis (PCA), and were used for the cluster analysis performed on two to eight clusters. Clustering Validity Indices (CVI) was used to determine the optimal number of clusters. Subsequently, each generated cluster was assessed for the potential risk of the occurrence of direct runoff, in five classes, on a scale from a moderate degree of risk to a high degree of risk. Six clusters were generated, which is the optimal number in terms of the CVI and their hydrological properties. In this case, 17% of the Czech Republic territory is assessed as lying within a high-risk area, 39% as lying within a medium-risk area, and 24% as lying within a below-average risk area in terms of the occurrence of direct runoff.

**Keywords:** hydrological response; cluster analyses; headwater catchment

## 1. Introduction

The mutual hydrological similarity of catchments, derived from the similarity of their response to a precipitation event, forms a basis for generalizing their hydrological links, and enables findings to be transferred between catchments. This approach enables catchments to be classified both in terms of their potential impacts on the environment and in terms of their vulnerability to hydrological extremes. Numerous authors have studied the similarities of catchments, their characteristics, and responses from various perspectives. For example, Wagener et al. [1] are motivated in their classification by long-term processes that affect the responses of catchments. A similar principle for the defining, classifying, and sharing the attributes of catchments has been adopted within the CAMELS data set [2,3]. The data sets are created for individual catchments to describe six main classes of attributes at catchment scale: topography, climate, streamflow, land cover, soil, and geology.

The "First-Order Catchments" (FOC), which are also referred to as headwater catchments or as upper catchments by some authors, form a specific group of catchments [4]. FOC are the backbone of the hydrographic network, and are the primary areas for capturing or mitigating flood-related damage. These catchments are often the most environmentally sensitive and the most rapidly evolving parts of many landscapes. They tend to be on the front line of environmental change, and pose the greatest challenges for those involved in land management, policy, and planning [5]. FOC also provide essential ecosystem goods and services for downstream areas [6].

The important parameters affecting the hydrological response from FOC are the properties of the soil and the land cover, morphological characteristics, and precipitation.

A large set of characteristics describes morphology characterization, including the average slope, hillslope lengths, and the mean topographic index [7]. The greatest risks in terms of discharges and factors that influence outflow from these catchments are caused by heavy rains and as a consequence of erosion. The precipitation-runoff relationships in the FOC are affected by the speed of the processes. Response to causal precipitation comes relatively quickly, and runoff is affected by intense sub-day precipitation, often of the convectional type. The most frequently-used tool for describing important attributes of precipitation are Intensity-Duration-Frequency (IDF) curves [8], which describe the relationships between rainfall intensity, rainfall duration, and frequency of repetition. The spatial distribution of the respective statistical attributes was studied on a worldwide scale in [9]. Kašpar [10] dealt with sub-daily precipitation, focusing on return period of precipitation using small river basins modelling. The hydrological response is also significantly affected by hyetograph on the local scale. Characteristic shapes of six hour precipitation were defined from long-term measuring with rain gauge and precipitation radar by Muller [11]. On the other hand, the advantage in very small river basins is greater homogeneity of precipitation events, and it is not necessary to use area reduction coefficients [12] to determine the causal precipitation.

Relatively few long-term observations have been made in these small catchments [13], or, for example, the GEOMON network [14]. Damage caused by flash floods and erosion, which are recorded by rescue services or are recorded by authorities can be used for localization, for example, monitoring of soil erosion [15] in the Czech Republic. Regulations on small streams and, potentially, in a catchment area, are more often proposed on basis of a hydrological model of design precipitation than real episodes. A simple method often used by design planners is the SCS-CN method [15], which has been continuously developed and tested [16,17]. By contrast, more complex, physically-based models, e.g., WEPP [18], SMODERP [19], TOPMODEL [20], and others, are used to describe the response of a catchment in a more precise manner.

The main potential risk of flood in the FOC is the occurrence of rapid runoff. Flood risk management based on a catchment scale approach is widely adopted [21,22] with principle being that run-off can be managed most effectively with a combination of measures in the area of the catchment and downstream flood protections [23]. Potential risks associated with the rapid occurrence of runoff in terms of the FOC characteristics that are more sloping have less infiltration capacity with less permeable land use and higher precipitation. The risk can also be assessed in terms of the potential threat to the infrastructure by flood or sediment flux [24].

In addition to the spatial delimitation of small headwater catchments, this article also introduces a set of parameters for classifying them in terms of the potential hydrological response. This article also presents a classification of catchments, and identifies the potential risks of the occurrence of rapid runoff.

The classification of streams is governed by various criteria. Classification based on the profiles of surface water bodies can be mentioned as an example [25]. Hierarchical categorization is also sometimes performed the other way round. First, the catchments of a major waterway are defined, and then they are further subdivided according to the importance of individual sub catchments.

This article is devoted to the derivation of small headwater catchments up to a catchment size of 5 km$^2$ in the Czech Republic territory. For catchments of this size, the use of hydrological models for determining the hydrological characteristics is assumed [26]. Catchments less than 5 km$^2$ in size can be considered so small that they can be viewed as homogeneous areas in terms of precipitation. The catchment classification adopted in the Czech Republic, which applies a hierarchical categorization at four levels, can serve as an example. The first level is represented by the major watercourses (the Elbe, Danube, and Odra Rivers), and these are subdivided at lower levels. Catchments differing in size from one km$^2$ to tens of km$^2$ are grouped in the most detailed fourth category. The classification of four level catchments is presented in [27]. The distribution of precipitation in the Czech Republic has been studied by [10].

## 2. Materials and Methods

Small headwater catchments (SHC) are the subcategory of FOC, and defined as catchments with a surface area of less than 5 km² [8] which, at the same time, do not have any tributaries according to the definition of a "first-order catchment" [4]. Small headwater catchments on the territory of the Czech Republic (see Figure 1) were identified according to the methodology described below. Parameters affecting the hydrological response, particularly with respect to the potential risks of rapid runoff due to extreme precipitations, were subsequently assigned to the catchment areas defined in this way. Short-term intensive precipitation events are crucial for potential flash flood risk factors in small catchments.

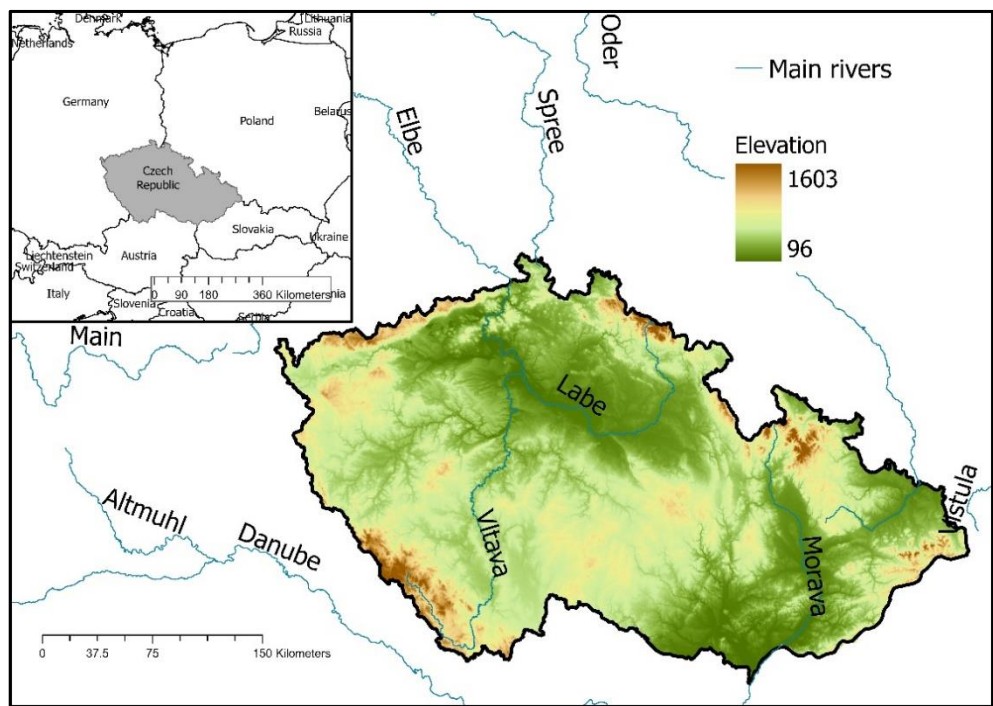

**Figure 1.** Study area of the Czech Republic with elevation model (CUZK—Zabaged®).

### 2.1. Definition of a Catchment

According to [26], SHCs are not only catchments with an area of at least 5 km², but also include all smaller catchments. This means, for example, that two catchments each with an area of 3 km² after the confluence do exceed 5 km², but are still regarded as two separate catchments falling within the SHC category. A total of six categories differing in size have been identified for defining an SHC; see Table 1. The classification of catchments into categories by their size enables comparisons to be made.

**Table 1.** SHC categories. Each category is characterized by area.

| Category | From km² | Up to km² |
|----------|----------|-----------|
| 005 | 0.3 | 0.7 |
| 010 | 0.7 | 1.3 |
| 020 | 1.7 | 2.3 |
| 030 | 2.7 | 3.3 |
| 040 | 3.5 | 4.5 |
| 050 | 4.5 | 5.5 |

Areas smaller than Category 005 can be considered as elementary runoff areas, and they are not assessed as areas of separate catchments. In total, three data sources were used to delimit SHCs: (a) the digital terrain model (DTM); (b) the streams shapes; and

(c) water reservoirs. The main input for the definition of SHC was ZABAGED® [28], with a resolution of 5 × 5 m. This DTM was generated on the basis of aerial LiDAR scanning. Due to anthropogenic interventions and changes in the landscape, the axes of streams have changed, and do not correspond with the natural flow paths generated on the digital terrain model itself. By contrast, the axes of streams that form part of ZABAGED® are based on measurements at more detailed scales, and reflect the present-day state of the stream network. In terms of flow direction, these paths are considered to be more accurate and more useful for generating SHCs than the DTM-based flow directions. While derived from the catchments, these stream axes must be included in the solution. The axes of the streams were projected onto DTM. The value of the terrain model pixels through which the axis of the stream passes was reduced by 100 m of altitude. The axes of the streams were therefore included in the generation of the flow direction, which was generated on the basis of DTM. In the next step, the potential places without runoff were discarded from the terrain model modified in this way, and subsequently an accumulation layer was created on the basis of the flow direction. The one-way flow direction tool was used to direct the flow (D8).

For each catchment category (see Table 1), the accumulation layer was reclassified so that the accumulation area values outside the group boundaries would have a NoData value, and the accumulation area corresponding to the given category would then be equal to 1.

The reclassified accumulation rasters generated for each size category were converted into vector lines. The lines were also intersected with the polygons of the water reservoirs included in the DIBAVOD (digital database of water management data). If the end part of the line was located in the water reservoir, the line was shortened to the point of entry of the flow line into the water reservoir. The point was identified by the intersection of the stream line with the polygon representing the respective water reservoir. The endpoints were determined for the modified lines in individual categories. The points identified in this way represent the end profiles of the SHC categories, to which the catchment boundaries were subsequently generated based on the model, taking into account the stream axes.

*2.2. Characteristics of Small Catchments*

The hydrological response of a small headwater catchment results from its morphological characteristics, soil properties, land use and the precipitation falling over the catchment. It can be assumed that the similar catchments will also have a similar hydrological response. The parameters for the classification of the SHC categories were therefore derived in terms of their potential hydrological response. The morphological characteristics were identified based on the DTM and the streams. These are, in particular, characteristics related to the elevation, the slopes, and the length of the flow paths. In addition, there are several shape coefficients.

The mean catchment width *b*:

$$b = \frac{A}{Fl_{max}} \tag{1}$$

The catchment shape factor *alfa*:

$$\alpha = \frac{A}{Fl_{max}^2} \tag{2}$$

The shape coefficient *gra* proposed by Gravelius [29]:

$$gra = \frac{P}{2 \cdot \sqrt{A \cdot \pi}} \tag{3}$$

All three shape coefficients describe the shape of a catchment. The Gravelius coefficient represents the difference in the shape of the catchment from a circle. The mean catchment width factor identifies the extent to which the shape of a catchment diverges from a square,

and the catchment shape factor alfa identifies the extent to which the shape of a catchment diverges from the power of maximal flow length path.

The stream network density (*SND*)parameter is a standard descriptive parameter. This parameter is calculated as the ratio of the total length of the streams to the catchment area:

$$SND = \frac{\sum L}{A} = \frac{total\ stream\_length}{area} \tag{4}$$

The lag time (Tlag) is a parameter which is affected by the morphology of the catchment, and affects the runoff pattern. Tlag is used to describe the unit hydrograph according to the SCS-CN method [15]. The Tlag value is calculated from the equation [30]

$$T_{lag} = Fl_{max*}{}^{0.8} \cdot \frac{(S+1)^{0.7}}{1900 \cdot \sqrt{Y}} \tag{5}$$

In Equations (1)–(5):
Tlag: lag time (hours)
$Fl_{max}$: length of the longest flow path (km)
$Fl_{max*}$: length of the longest flow path length (feet)
$\sum L$: total stream length (km)
Y: catchment mean slope [-]
S*: maximum potential retention height marked (inch)

The direct runoff potential of a catchment is described by the mean value CN. CN integrates information on land use (LU) and soil infiltration properties. In the Czech Republic, CN values were derived within the Strategy for Protection against Negative Impacts of Floods and Erosion Phenomena by Nature-Friendly Measures in the Czech Republic [31].

The final group of parameters consists of precipitation data. As short-term precipitation is the dominant source of runoff in small headwater catchments, six-hour design totals derived from rainfall radars with spatial resolution of $1 \times 1$ km were applied [10,32]. These data are available at rain.fsv.cvut.cz, and the mean value for each SHC was calculated and used for this study.

All monitored parameters are listed in Table 2. These values describe the mean value, the variance, or maximum value, according to the type of parameter.

In total, there were 28 parameters, which were subsequently tested in all catchment size categories for mutual dependence using regression analysis. The objective was to obtain a set of independent parameters, and to classify the catchments into groups according to their similarity, using cluster analysis.

**Table 2.** List of parameters that enter the SHC classification. Names of each parameter included type of parameter; where relevant, it also contains a description of the type of statistics. The statistics (standard deviation (STD), mean, maximum) are calculated in GIS. Parameters 1–12 are derived from DTM, or in combination with the stream axes, where the slope characteristics (15–18) were identified in the original DTM layer before the stream receded. Parameters 20–24 were obtained from Equations (1)–(4).

| No. | Group | Name and Description | No. | Group | Name and Description |
|-----|-------|----------------------|-----|-------|----------------------|
| 1 | Basic parameters | Perimeter | 15 | Slope parameters | Slope_mean—average slope |
| 2 | | Area | 16 | | Slope_STD—standard deviation |
| 3 | | Elevation mean—average catchment elevation | 17 | | Slope_stream_mean–average slope of the streams |
| 4 | | Elevation STD—deviation in elevation describing the flatness of the catchment | 18 | | Slope_stream_STD—deviation in the slope of the streams |

<div align="center">**Table 2.** *Cont.*</div>

| No. | Group | Name and Description | No. | Group | Name and Description |
|---|---|---|---|---|---|
| 5 | Flow accumulation parameters | Fl_acc_mean—average flow accumulation | 19 | Morphological parameters | Medium width—mean catchment width |
| 6 | | Fl_acc_STD—deviation in flow accumulation | 20 | | Shape coefficient Alpha |
| 7 | Flow length parameters | Fl_len_max—maximum flow path length | 21 | | Gravelius coefficient—shape coefficient |
| 8 | | Fl_len_mean—average flow path length | 22 | | Tlag—lag time |
| 9 | | Fl_len_STD—standard deviation in the length of the flow path | 23 | Stream parameters | Total stream length—total length of the streams |
| 10 | | Fl_len_noStream_max—maximum length of the surface runoff flow path | 24 | | SND—stream network density |
| 11 | | Fl_len_noStream_mean—average length of the surface runoff flow path | 25 | Mean six-hour design precipitation | P2—rainfall with returnPeriod (RP) 2 yr |
| 12 | | Fl_len_noStream_STD—standard deviation in the surface runoff flow path | 26 | | P10—rainfall with RP 10 yr |
| 13 | SCS-CN parameters | CN_mean—average CN number for the whole catchment | 27 | | P20—rainfall with RP 20 yr |
| 14 | | CN_STD—standard deviation | 28 | | P100—rainfall RP period 100 yr |

The delimitation of small catchments, and the assignment and calculation of their characteristics from DTM and the CSC-CN of the catchment were processed using the ESRI environment tool (ArcGIS and ArcGIS Pro).

Descriptive statistics, regression analysis, and principal components analysis (PCA) [33,34] were used for to exclude correlating parameters, and to narrow down the selection of the number of variables for clustering analyse. Cluster analysis was performed using the K-mean method. Clustering validity indices (CVI) [35] were used to determine the optimal number of clusters. Statistical and clustering analyses were performed in the R environment.

## 3. Results

### 3.1. SHC Delimitation

Basic data on SHC derived according to the methodology described above are presented in Table 3. As the SHC categories are always derived separately, the resulting catchments overlap between the categories; a smaller resulting catchment may be part of a larger catchment in the higher categories. Therefore, in addition to the categories described above, a group of catchments was generated, in which only the largest catchments were represented. The nested catchments were eliminated. In this way, catchments smaller than 5 km$^2$ situated in the monitored area in the Czech Republic have been preserved. The resulting group of catchments generated in this way is from now on referred to as the "Set of the Largest Catchments" (SoLC), and is also listed in Table 3. For clarity, and to give an idea of the representation of each size category in the resulting SoLC group, the table contains data on the number of elements in the respective category that form part of the SoLC.

**Table 3.** Number of catchments and the total area of catchments in each category. The first two columns describe the number of elements in the respective category, and the last two columns indicates the number of elements in SoLC.

| Category | Number of Elements | Total Area (km$^2$) | Representation of Elements in the SoLC | | | |
|---|---|---|---|---|---|---|
| | | | Number | % | km$^2$ | % |
| 005 | 72,621 | 37,632 | 16,894 | 23 | 7727 | 12 |
| 010 | 31,287 | 33,046 | 10,907 | 35 | 11,038 | 18 |
| 020 | 11,560 | 24,179 | 3938 | 34 | 8051 | 13 |
| 030 | 6530 | 20,289 | 2187 | 33 | 6655 | 11 |
| 040 | 5431 | 22,610 | 2271 | 42 | 9086 | 14 |
| 050 | 3957 | 20,479 | 3957 | 100 | 20,478 | 32 |
| SoLC | 40,154 | 63,031 | | | | |

### 3.2. Choice of Parameters

The parameters for each catchment in all size categories were derived according to Table 2. For the purposes of cluster analysis, representative and independent parameters were sought in the first step. Dependent parameters must be discarded. The search for the degree of agreement between the monitored parameters was performed both for individual categories (including SoLC) and for all the catchments together. In terms of the groups of dependent parameters, the individual categories do not differ from each other. Therefore, the relationships between the monitored parameters are similar for all size categories. The visualized parameter agreement is shown in Figure 2.

Figure 2 shows the parameters that are mutually correlated in the marked groups A–D. Principal Component Analysis (PCA) was used for selection of representative parameters. PCA was used for (i) all parameters in first round and (ii) by the groups (A–D) in second.

(i) It was found that the first two principal components of PCA of all data do not explain even 50% of the variability. The first five components: 75% of the variability; the remaining six components: 80% of the variability (Figure 3).

The contribution of individual parameters, especially in the case of the first two substitute variables, does not exceed 9% and 13%, respectively (see Figure 4). PCA analysis did not provide a clear choice of explanatory variables (see Figure 4).

(ii) The first two dimensions of the PCA analysis explain for groups A–D: 80% of the already embodied 98 variability. The effect of the first two components is given in more detail using biplots in Appendix C. The PCA analysis did not show any significant factors that should not be overlooked. The final selection of representative parameters was chosen by combining the significance of the element in the PCA analysis, with regard to the usability of the selected parameters in hydrological modelling.

The six parameters were selected. Group C is represented by CN_mean. Slope_mean can also characterise group C. Moreover, CN_mean represents the retention capacity. Precipitation data have a separate group—group D. The parameter Tag from group A was chosen in the same way. This parameter interpreted the input to hydrological models more than the slightly better explanatory variables Fl_len_max or Fl_len_STD. A close correlation ($R^2 > 90$) was found between all precipitation data. Rainfall with a return period of 20 years (P20) was chosen with regard to the possible hydrological response. With shorter repetition times, thus precipitation totals, the hydrological model may not lead to runoff, a repetition time of 100 years is too extreme, and the time series are relatively short to derive this repetition. Three parameters from group B were selected. Parameters SND and Alpha, around which subgroups are formed, together correlate with the parameter Fl_len_noStream_STD. All three of these parameters are explanatory variables of the third PCA component. Fl_len_noStream_STD explains (20%), SND (12%), and Alpha (7%) of this component. SND and Alpha do not correlate with each other, and together have the same explanatory weight as Fl_len_noStream_STD. SND and Alpha are direct value of the parameter and not of the Standard Deviation Parameter, and are more relevant to assign a risk level to these parameters.

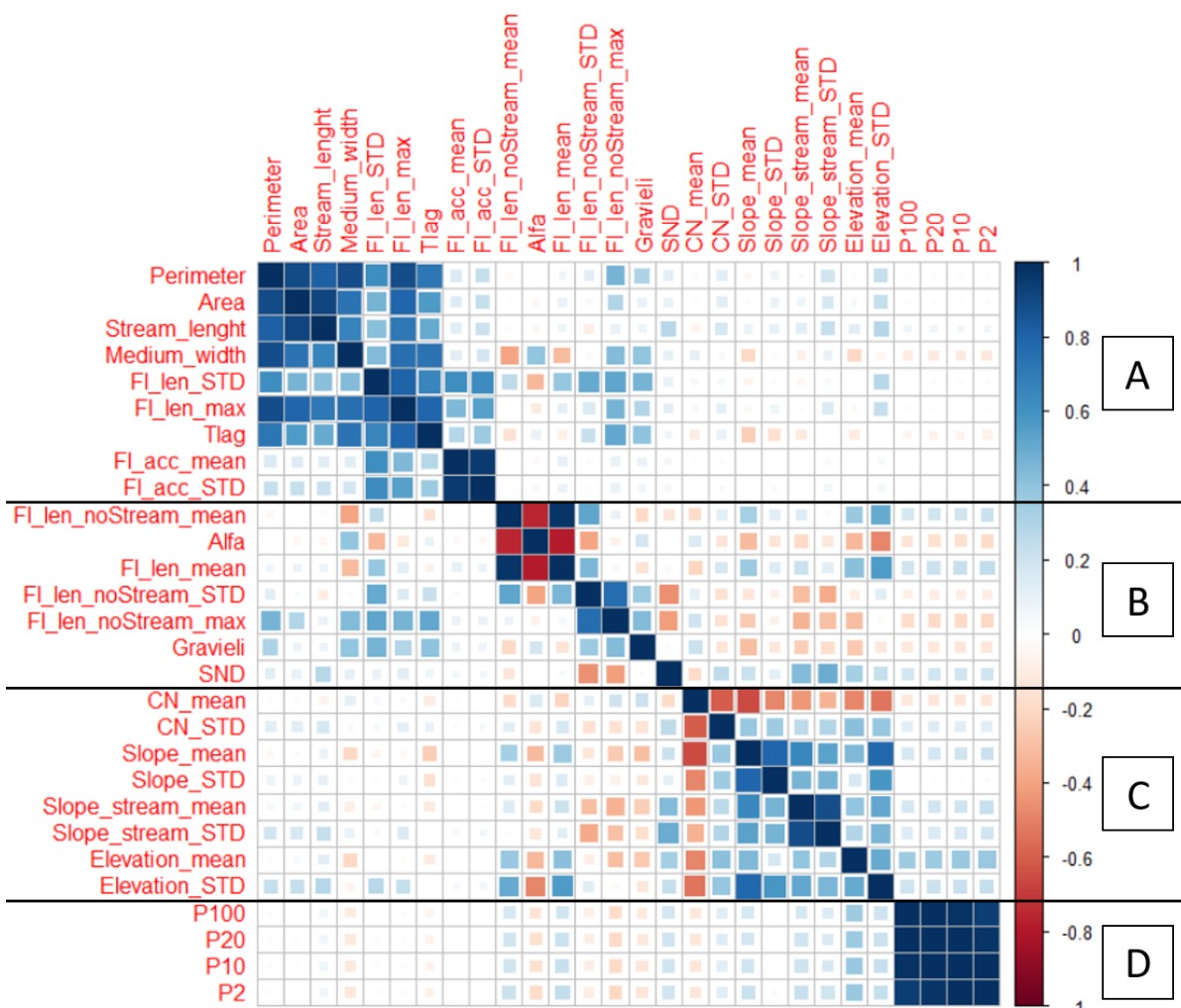

**Figure 2.** Visualization of the correlation coefficients between individual parameters for all catchments regardless of size category. A negative correlation is shown in red, and a positive correlation is shown in blue. The stronger the correlation between two parameters, the darker and larger the symbol. Parameters in correlation are grouped. The main groups (**A–D**) are separated by a black line. A description of the parameters is provided in Table 2.

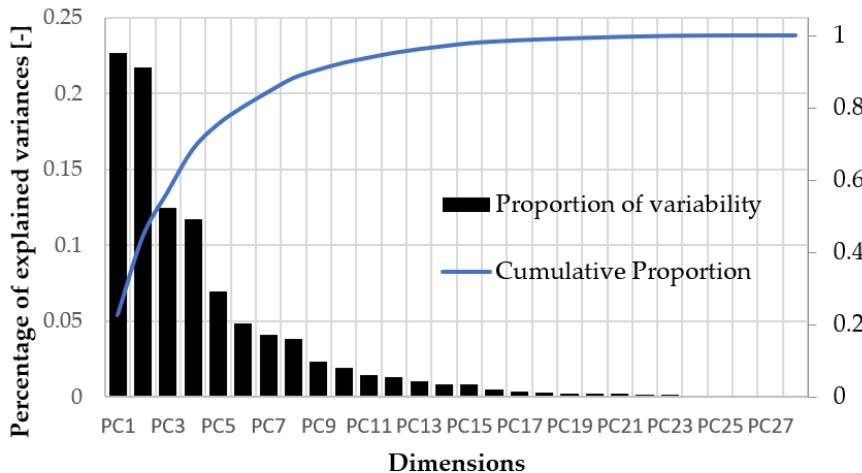

**Figure 3.** Percentage of explained variances with PCA, individual proportion (black) and cumulative (blue).

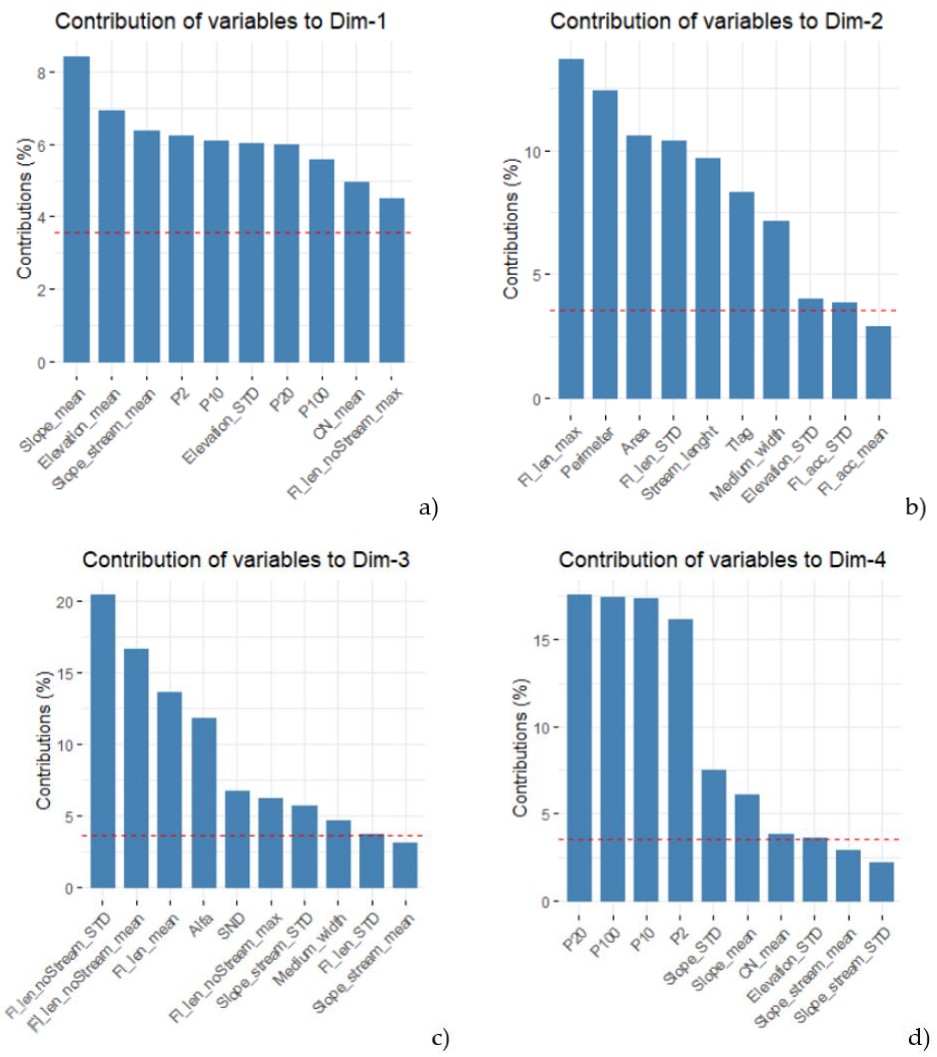

**Figure 4.** Contribution of variables for first four dimensions of PCA for all parameters. (**a**) to first, (**b**) second, (**c**) third, (**d**) fourth substitute variables.

Five parameters that can be considered independent and sufficiently representative (groups A–D) are as follows:

- Precipitation P20—there is a significant correlation between individual values of six-hour precipitation events due to the derivation of these data;
- The CN mean, which represents a group of several other parameters. The CN value shows agreement both with the slope and with the altitude;
- Lag time (Tlag)—this parameter characterizes the shape of the runoff hydrograph, and therefore the peak flow volume;
- Stream Network Density (SND)—this parameter represents the density of the streams and amount of the stream network;
- Shape coefficient (Alpha)—This parameter incorporates the characteristics of the flow path length and of the catchment shape.

### 3.3. Distribution of Parameters

In order to classify the catchments into groups in terms of their potential response, it is necessary to compare the distribution of the classification parameters among the individual catchment categories. If the selected classification parameters were to show a different distribution for each group of catchments, it would imply that different size categories manifested a different type of hydrological response to a precipitation event. The objective

is, therefore, to compare the differences between the respective size categories. The distribution of the five parameters selected for the individual size categories is presented in Figure 5. Visualization in the form of histograms has been chosen, showing the distribution of values and box plots, displaying the median value, the interquartile range, and outliers. The number of classes is the same for all histograms. The numbers of elements in the displayed histograms are normalized so that the distribution of the catchment representation in individual classes can be compared.

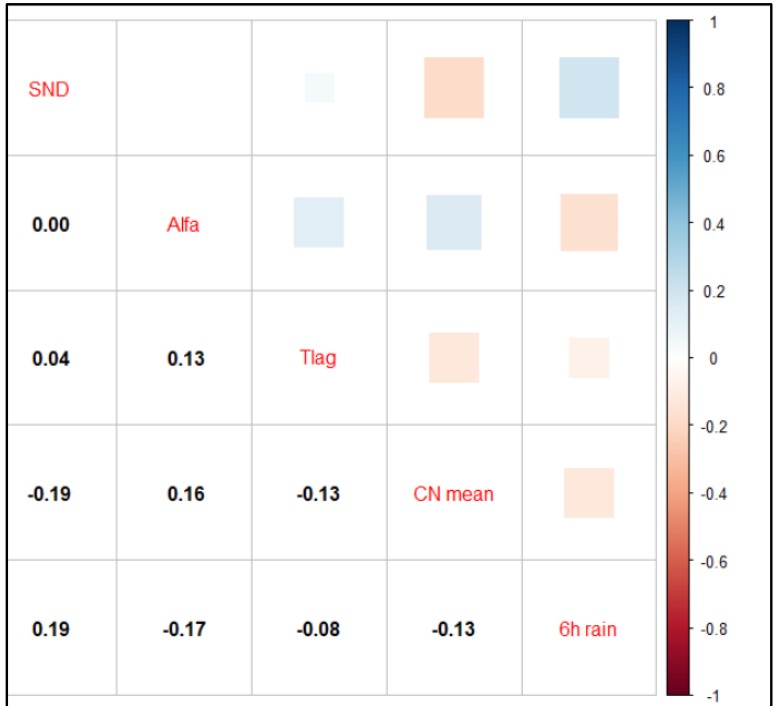

**Figure 5.** Correlation of five selected parameters. The correlation coefficients are numerically shown below the diagonal, and the correlations are visually presented above the diagonal, as in Figure 2. The selected parameters can be regarded as independent variables.

As the results in Figure 6 clearly show, the differences between the catchment size categories are not significant. There is an obvious difference in the SND specification for the 0.5 and 1 km catchment size, where catchments without a hydrographic network occur to a greater extent. In the other size categories, catchments without a hydrographic network occur significantly less often. The second clear difference is in the Tlag value, which also partially includes the catchment size; the concentration times are generally higher in the larger categories.

A cluster analysis for each category separately would not provide information significantly different from SoLC in terms of runoff potential. Cluster analysis was therefore performed for SoLC. Each catchment size categories are represented at least 20% in SoLC.

Cluster analysis (K-mean) was performed by five representative parameters for two to eight clusters, with 25 initial training points. The formation of groups of catchments is shown in Figure 7. Groups are marked with letters. If a group is generated only by separation from a previously formed group, a numeric designation is added.

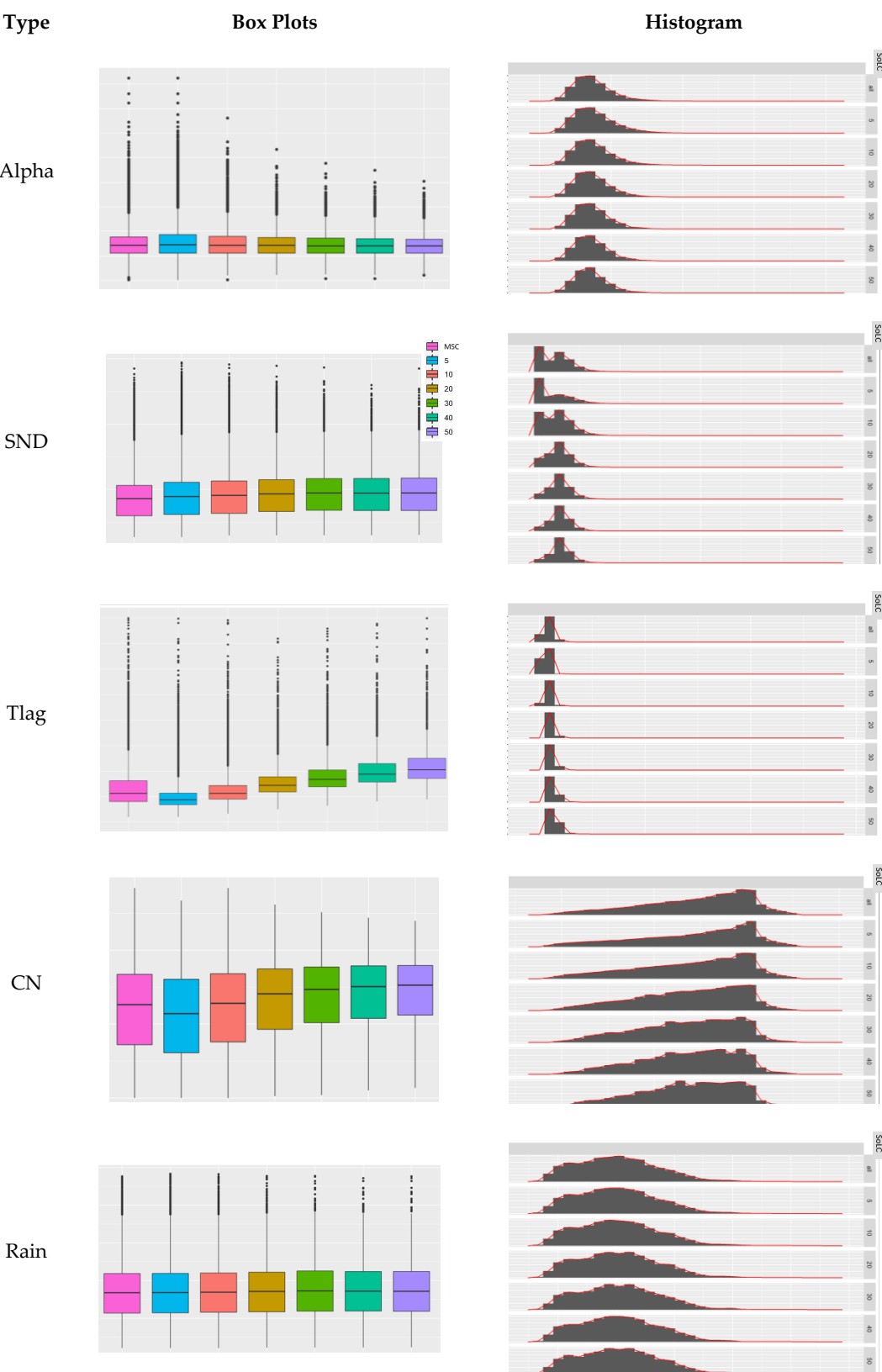

**Figure 6.** The left column displays the box plots showing the median value, the interquartile range, and the 25 or 75 quartiles of the values. The right column displays the frequency histograms in each class. The values in the histograms are normalized to achieve a comparable number of values in each catchment size category.

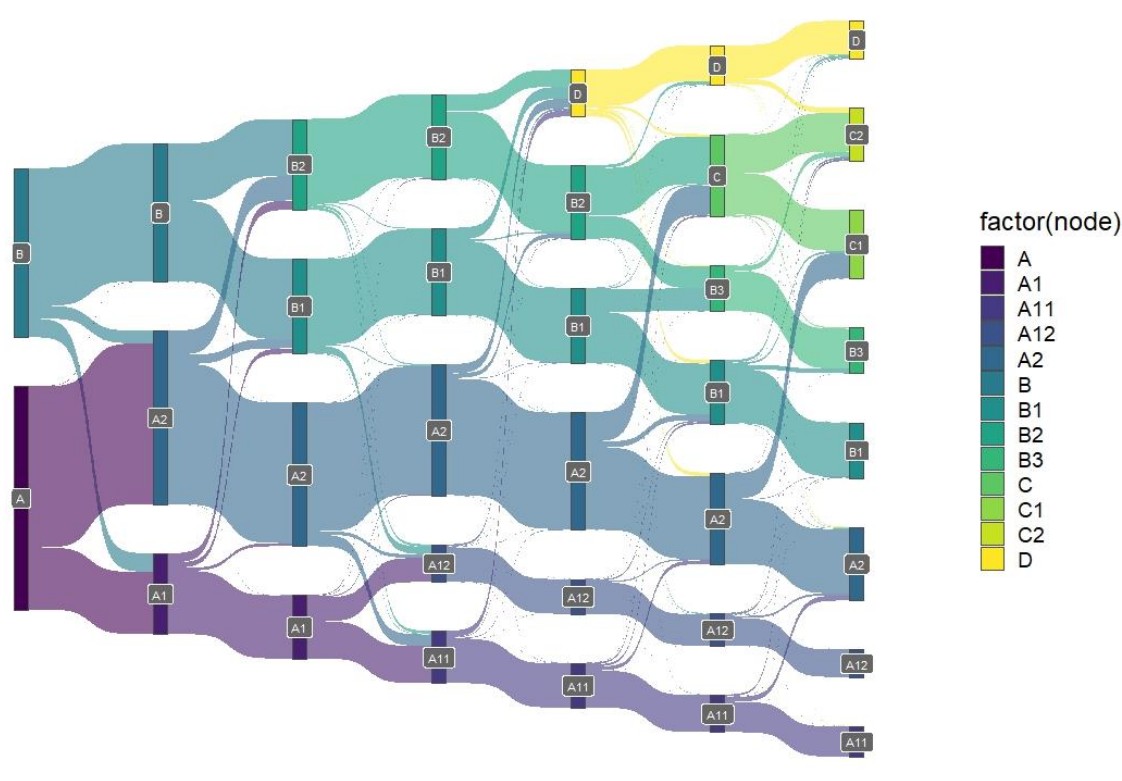

F2_Clusters F3_Clusters F4_Clusters F5_Clusters F6_Clusters F7_Clusters F8_Clusters

**Figure 7.** Sankey diagram of the development of clusters of SoLC. The widths of the bands are proportional to number of elements. The diagram also shows transfer of individual catchments and regrouping together with the growing number of clusters. The basic subdivision is already evident when two main groups (A, B) are formed. The groups generated up to five clusters, which are subdivisions of A or B. Group D, which is a combination of previously formed groups, is generated in the six clusters step. Group C arises when the number of clusters reaches seven and eight as combinations of Groups A2 and B2.

The groups generated during gradual clustering are characterized below. Geographical clustering is shown in Figure 8. The centroids of each cluster for the built clusters are listed in Appendix A.

- Two Clusters—When the first two clusters are formed, Group A is generated, which is characterized by a somewhat higher CN number with lower precipitation volumes. Group B is characterized by higher precipitation volumes and a higher CN value (Figure 8a);
- Three Clusters—Group A is divided mainly in terms of the shape characteristics of the catchment, the density of the stream network, and the lag time (Tlag) (Figure 8b);
- Four Clusters—Group B1, which is characterized by lower precipitation volumes while maintaining a lower CN value, and, by contrast, Group B2 with higher precipitation totals and, at the same time, a higher CN value, are separated from Group B (Figure 8c);
- Five Clusters—Group A1 is predominantly divided on the basis of the lag time. The resulting group A12 is characterized by a significant Tlag time, while the initial characteristics of Group A1 are fairly preserved in Group A11. Group A11 and group A12 defined in this way are preserved even after the catchments have been subdivided into more clusters (Figure 8d);
- Six Clusters—A completely new Group D is generated, which is characterized by a relatively high SND value and, at the same time, relatively low precipitation totals, while maintaining a relatively high CN value. Group D, generated in this way, is preserved even after the catchments have been subdivided into more clusters (Figure 8e);

- Seven Clusters—Group B2, which is characterized by relatively high precipitation volumes, has been noticeably divided. Together with some of the catchments of Group A2, it generates a new Group C, which is characterized by higher precipitation totals and, at the same time, higher CN values. Some of the catchments of initial Group B2 and some of the catchments of Group B1 generate Group B3, which retains parameters similar to those of initial Group B2. However, the number of catchments in initial Group B2 is so small that the group is renamed B3 (Figure 8f);
- Eight Clusters—The newly generated Group C is regrouped into Groups C1 and C2. The newly arising Group C1 is also supplemented by some of the catchments of Group A2 which, similar to the initial Group C, are characterized by higher precipitation volumes and the CN value. Consequently, Group C 1 differs from Group C2 by the difference in the SND and Alpha parameters (Figure 8g).

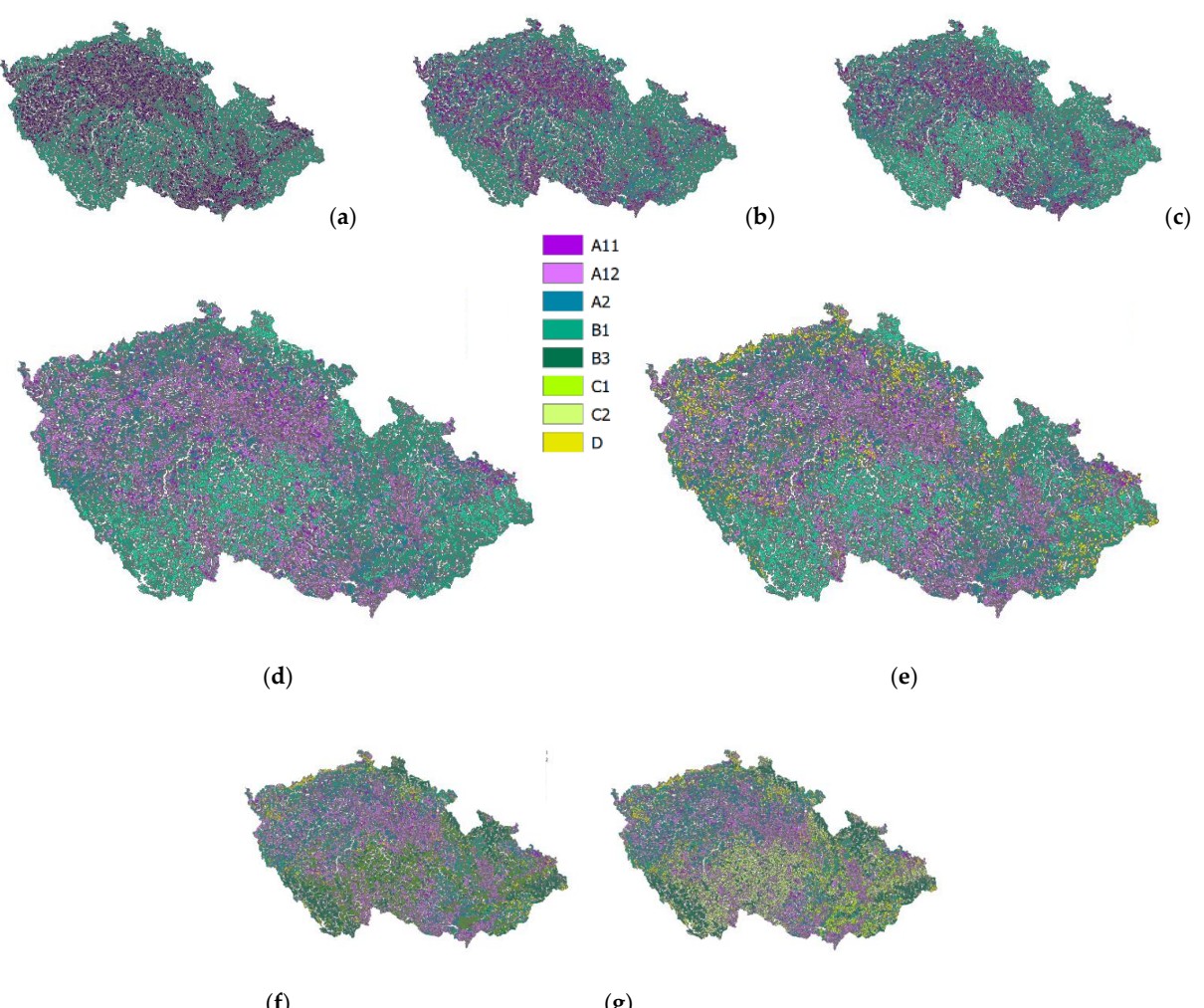

**Figure 8.** Geographical representation of the development of the groups during the formation of two to eight (**a**–**g**) clusters.

The gradually generated groups of catchments are characterized by the mean values of the five selected parameters.

The Davies-Bouldin index was used to determine the optimal number of clusters using Clustering Validity Indices (CVI) (Figure 9). The optimal number of clusters from this is five. A suitable number of clusters is six, from which completely new class D arises (described in more detail in discussion).

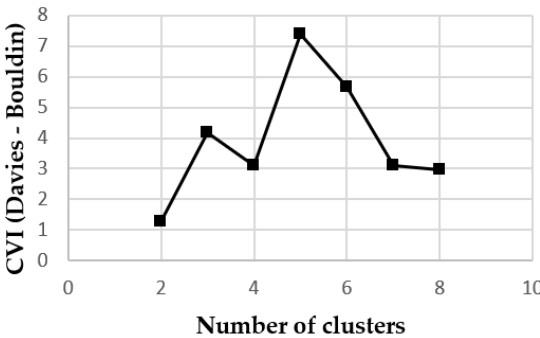

**Figure 9.** CVI (Davies-Bouldin).

The generated clusters of the catchments are further classified according to the potential risks of the occurrence of rapid runoff. In terms of the effect of the parameters on the risks associated with the rapid occurrence of runoff, the following holds for individual parameters:

- SND—the higher the value, the denser the network of permanent streams, the more likely the potential runoff is to be concentrated in these paths, where runoff is expected. A higher value means a lower level of risk;
- Tlag—the greater the lag time, the lower the anticipated peak flows;
- Alpha—the more complex the shape of the catchment is, the longer the flow paths and thus the lower the culmination.
- CN mean—the lower the CN mean value, the greater the retention capacity in the catchment, and the lower the risk of a potential threat;
- 6 h rain—the more intensive the rain is, the higher the risk of a potential runoff response.

The mean value, which is considered as a medium level of risk, was calculated for individual parameters in the SoLC category. The degree of risk was identified for the individual parameters relative to this mean value of the respective parameter. For each parameter value corresponding to the centroid of the individual clusters, the quotient was identified using this mean value, which yielded the degree of risk of each parameter in the cluster. Combinations of five parameters where a negative assessment prevails were then considered risky, and vice versa. The overall degree of risk was divided into five categories, from low risk to high risk, as described below.

- Low risk—the combination of the parameters of a potential runoff response implies a small risk. in terms of rapid direct runoff affecting the catchment. These areas appear to be unproblematic in terms of a rapid response, and there is no need to implement any measures;
- Decreased risk—the combined parameters of a potential runoff response imply a rather small risk in terms of rapid runoff affecting the catchment. These areas appear to be unproblematic in terms of a rapid response, and there is low need to implement any measures;
- Medium risk—the combined parameters of a potential runoff response are average, and an average degree of risk is assumed in terms of rapid runoff affecting the catchment;
- Increased risk—the combined parameters of a potential runoff response imply a rather higher degree of risk in terms of rapid runoff affecting the catchment;
- High risk—the combined parameters of a potential runoff response imply a great risk in terms of rapid runoff affecting the catchment. In these areas, a more detailed survey and more detailed monitoring of potential negative impacts of rapid runoff need to be carried out.

The parameter values for identifying the degree of risk are listed in Table 4.

**Table 4.** Values of individual parameters used to express the degree of risk in relation to the mean values of the parameters.

| Risk Coefficient | Low Risk <0.85 | Decreased Risk <0.95 | Medium Risk <1.05 | Increased Risk <1.15 | High Risk >1.15 |
|---|---|---|---|---|---|
| SND | 1.19 | 1.09 | 1.03 | 0.98 | 0.88 |
| Tlag | 3.75 | 3.43 | 3.26 | 3.10 | 2.77 |
| Alpha | 4.29 | 3.92 | 3.73 | 3.55 | 3.17 |
| CN mean | 79.20 | 72.31 | 68.87 | 65.42 | 58.54 |
| 6 h rain | 42.75 | 47.78 | 50.30 | 52.81 | 57.84 |

The classification of the catchment groups formed by cluster analysis according to the degree of risk is presented in Appendix A. Groups with two–eight clusters are included. The classification of the catchment groups produced by cluster analysis, according to the degree of risk of the developing clusters, is presented in Appendix B. The geographical representation of the degree of risk of six clusters is shown in Figure 10. Group A2 forms a high-risk group, Groups A11, B1, and B2 form a medium risk group, and Groups A12 and D form a group with a "decreased level of risk".

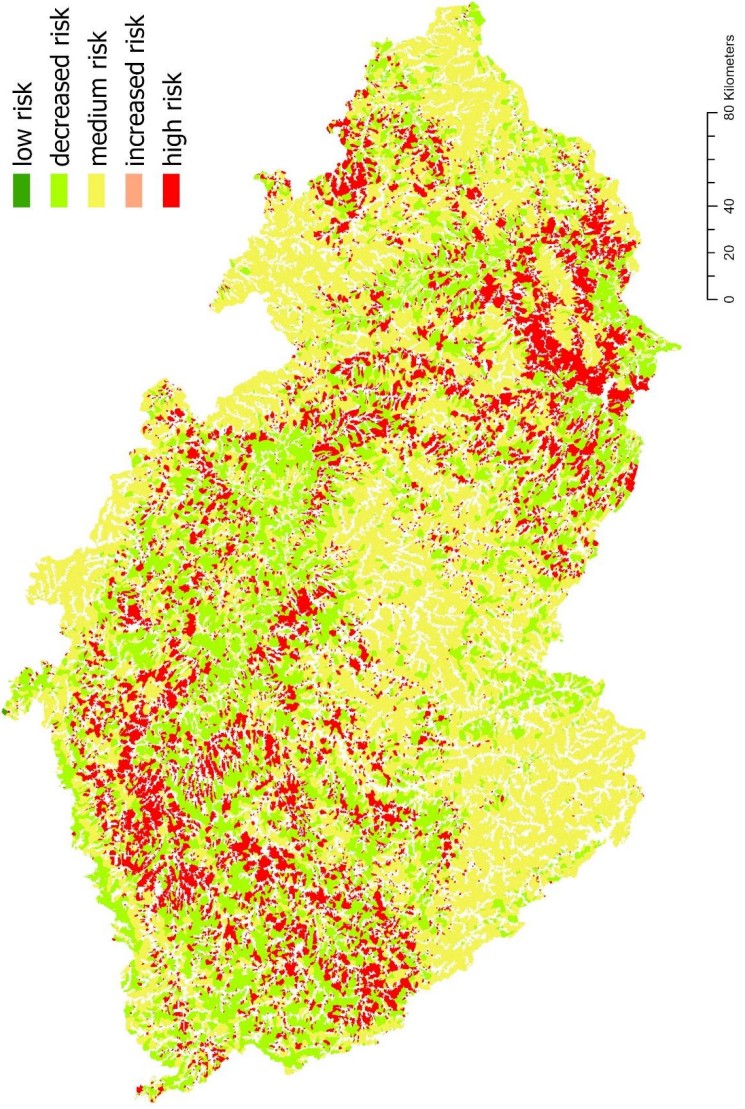

**Figure 10.** Degree of risk of the Small Headwater Catchments for case of six clusters—optimal count.

## 4. Discussion

Catchment classifications are more frequently performed in experimental hydrology. In the expanding CAMELS database, the catchments are also classified. In the case of CAMELS, there is a longer list of parameters [2]. Unlike the selected parameters presented here, the list of parameters is extended to include the hydrological data of long-term balances and parameters that have a more significant effect on long-term runoff and on other components of the balance. In most cases, larger catchments are investigated. Long-term observation campaigns are conducted considerably less frequently in small catchments than in larger ones.

Input data with different spatial resolutions were used to generate the boundaries of the catchments and their properties. The catchment boundaries were delimited on the basis of a terrain model with a resolution of $5 \times 5$ m, which is sufficiently detailed even for the delimitation of small catchments in size category 005. Based on the terrain model, other morphological characteristics were derived with the same resolution. If the D8 method were used with lower resolution, the generation of the catchment boundaries could be affected, mainly in the smallest category.

Within the investigated catchment size categories from 0.5 to 5 km$^2$, a significantly higher number of catchments are grouped in the first category (005). The parameters entering the cluster analysis do not differ significantly in terms of the distribution of values between the categories. The smaller catchments are also form part of larger categories, and together they form SoLC (Set of Largest Catchments), where at least 20% of the number of catchments in each category is represented. The total area of the upper catchments included in SoLC is 63 thousand km$^2$, which amounts to 80% of the area of the Czech Republic (78 thousand km$^2$).

A total of 28 parameters were considered to affect hydrological response. Some of the parameters create groups of mutually corelated parameters. PCA provided only a partial answer to the selection of the most suitable parameters from whole dataset. That the selection of representative elements is based in combination of PCA, pair correlations of parameters, and usability of the parameter in hydrological models. Selection of the parameters is discussed in the results, which is most logical. Slope parameters, stream slope and the altitude are considered to be in relation to CN. This correlation corresponds to the land use in mountainous areas, which are mostly more sloping, and are mostly forested. Another group of mutually correlated parameters are the areal parameters of the catchments (size, area, flow path length, runoff accumulation), which that are correlated with Tlag. SND and Alpha correlate with standard deviation of sheet flow length. However, this pair (SND and Alpha) is more often used in hydrological models. Precipitation events forming a separate group of parameters are mutually strongly correlated. However, they do not correlate significantly with the other parameters.

In SHC, short-term rains, in particular, are crucial in terms of response and, potentially, in terms of the risk of increased flows. A specific flood event and a potential threat occur when the current conditions in the catchment are combined with the course of the precipitation. Short-term torrential rains are events particularly difficult to forecast. However, two differently classified catchments that are in the same initial condition, and are subjected to the same precipitation will show a different response to similar precipitation events.

The subsequent cluster analysis of the catchments in terms of their hydrological response risk shows that, based on the selected parameters. SHC (small headwater catchments) can be subdivided and can split into two groups. In these groups, Categories A2 and B2 are gradually separated, and their parameters cause them to fall into the group with a risk of an increased runoff threat due to torrential rains. CVI and Davies-Bouldin [35] were used to determine the optimal number of clusters. Other indices were tested (Calinski Harabasz and Ratkowsky Lance) [35] without clear conclusion about the most appropriate number of clusters. The optimal number of clusters from Davies-Bouldin is five. Groups C and D arising from A and B previously generated sub-groups in six and more

clusters. Group D that arises in six clusters has the main characteristic of high SND value—catchment with high stream density (Appendix A), and should be preserved in the final grouping. Group C is later separated from two groups (A2 and B2) high risk catchments. The formation of two groups, C1 and C2, from Group C and partially from Group A2, with a total of eight groups, no longer provides new information in terms of potential threats.

In relation to the hydrological response type and the possible level of risk, the view of the final number of clusters is determined not only by the CVI result, but also by values of centroids. Statistical processing did not show any important parameter that would not be used in the analysis. A comparison with the recorded events [36] can be used in feature work to verify the risk classification of SHC. Classification of catchments according to parameters has a practical impact for potential prioritization in terms of the implementation of protective measures.

## 5. Conclusions

The Czech Republic is an example of a country where headwater catchments form a significant part of the territory. In the case of SoLC, 80% of the Czech Republic is covered in this category. SHC is the space for the primary accumulation of rainwater. At the same time, SHC tend to be more affected by rapid runoff, which, in turn, reduces the availability of water in the 'headwater catchment' area. The SHC classification presented here specifies the degree of threat, and reveals the probable hydrological response of each cluster of catchments. Regarding the first five clusters, the primary subdivision into two groups, A and B that arise early as the first two clusters. Within Group A and Group B, two subgroups are gradually formed, which are moderately high-risk groups. A sufficiently explanatory classification of SHC uses seven clusters, where a very low-risk group (D) is generated from the elements of Groups A and B; and Group C, with a high degree of risk, is separated from Groups A2 and B2. With six clusters covering the area of the Czech Republic, this approach places 17% of the territory in the high-risk category, 39% in the medium-risk category, and 24% in the below-average risk category. A total of 20% of the territory of the Czech Republic is not assessed, as it does not fall into the SHC. Within the Czech Republic, the agriculturally exploited areas of south Moravia, the Bohemian-Moravian Highlands, and north-west Bohemia can be considered as regions with a more significant degree of risk. The medium risk areas are mainly mountainous regions, with the exception of the Krušné Mountains, which are a low-risk area. The lower risk areas include the foothills, with the exception of the Podorlicko region, the Elbe River lowlands, the Brdy Mountains and West Bohemia.

The classification of small headwater catchments in terms of the threat by potential torrential rains is one of the perspectives. Another use of the spatial delimitation of these catchments could be for a subsequent classification, for example, in terms of the availability of water for irrigation or in terms of the application of other adaptation measures due to anticipated changes in climate.

**Funding:** This research was supported by the Czech Ministry of Agriculture (grant No. QK1910029).

**Informed Consent Statement:** Not applicable.

**Data Availability Statement:** The data is available in the web application rain.fsv.cvut.cz (accessed on 26 October 2021).

**Conflicts of Interest:** The author declares no conflict of interest and the funders had no role in the design of the study; in the collection, analyses, or interpretation of data; in the writing of the manuscript, or in the decision to publish the results.

# Appendix A

**Table A1.** A table presenting the development of the parameter values in the cluster centroids during the progressive formation of two to eight clusters. The table also shows the number of elements in the respective in each group (Count) of the total number of SHCs in SoLC, the in SoLC. The value is also expressed as a percentage.

| Number of Clusters | Group | Count | % of SoLC | SND | Tlag | Alpha | CN Mean | 6 h Rain |
|---|---|---|---|---|---|---|---|---|
| 2 | A | 22,858 | 56.9 | 0.62 | 3.04 | 4.07 | 75.22 | 47.59 |
| 2 | B | 17,294 | 43.1 | 1.59 | 3.56 | 3.28 | 60.47 | 53.87 |
| 3 | A1 | 8239 | 20.5 | 1.15 | 4.86 | 5.50 | 72.62 | 47.86 |
| 3 | A2 | 17,783 | 44.3 | 0.54 | 2.50 | 3.41 | 75.05 | 48.24 |
| 3 | B | 14,130 | 35.2 | 1.59 | 3.29 | 3.10 | 58.90 | 54.31 |
| 4 | A1 | 6484 | 16.1 | 1.01 | 5.04 | 5.72 | 73.25 | 47.17 |
| 4 | A2 | 14,689 | 36.6 | 0.45 | 2.53 | 3.41 | 75.71 | 47.01 |
| 4 | B1 | 9712 | 24.2 | 1.30 | 3.48 | 3.12 | 53.91 | 49.80 |
| 4 | B2 | 9267 | 23.1 | 1.70 | 2.95 | 3.48 | 70.63 | 58.21 |
| 5 | A11 | 5326 | 13.3 | 0.91 | 3.09 | 6.18 | 74.80 | 47.47 |
| 5 | A12 | 3826 | 9.5 | 1.07 | 7.07 | 4.10 | 69.02 | 47.48 |
| 5 | A2 | 13,427 | 33.4 | 0.45 | 2.51 | 3.26 | 75.65 | 47.16 |
| 5 | B1 | 8933 | 22.2 | 1.32 | 3.20 | 3.11 | 53.48 | 50.02 |
| 5 | B2 | 8640 | 21.5 | 1.71 | 2.92 | 3.44 | 70.51 | 58.44 |
| 6 | A11 | 4578 | 11.4 | 0.66 | 3.12 | 6.33 | 74.66 | 47.66 |
| 6 | A12 | 3570 | 8.9 | 1.01 | 7.19 | 4.11 | 69.03 | 47.52 |
| 6 | A2 | 12,031 | 30.0 | 0.35 | 2.52 | 3.26 | 75.81 | 46.99 |
| 6 | B1 | 7718 | 19.2 | 1.16 | 3.27 | 3.06 | 52.29 | 50.57 |
| 6 | B2 | 7526 | 18.7 | 1.32 | 2.95 | 3.38 | 70.21 | 59.83 |
| 6 | D | 4729 | 11.8 | 2.49 | 2.83 | 3.78 | 70.38 | 47.72 |
| 7 | A11 | 3829 | 9.5 | 0.70 | 3.21 | 6.56 | 75.26 | 48.00 |
| 7 | A12 | 3333 | 8.3 | 1.03 | 7.33 | 4.13 | 69.22 | 48.12 |
| 7 | A2 | 9327 | 23.2 | 0.40 | 2.66 | 3.39 | 76.36 | 44.77 |
| 7 | B1 | 6501 | 16.2 | 0.94 | 3.12 | 3.40 | 54.19 | 46.95 |
| 7 | B3 | 4668 | 11.6 | 1.64 | 3.44 | 2.72 | 56.11 | 60.04 |
| 7 | C | 8424 | 21.0 | 0.83 | 2.57 | 3.38 | 75.06 | 56.06 |
| 7 | D | 4070 | 10.1 | 2.67 | 2.83 | 3.94 | 70.63 | 49.15 |
| 8 | A11 | 3127 | 7.8 | 0.72 | 3.23 | 6.84 | 75.07 | 48.07 |
| 8 | A12 | 2817 | 7.0 | 1.00 | 7.67 | 4.16 | 68.66 | 47.96 |
| 8 | A2 | 7515 | 18.7 | 0.49 | 2.89 | 3.82 | 76.35 | 43.39 |
| 8 | B1 | 5683 | 14.2 | 0.88 | 3.16 | 3.50 | 53.98 | 46.36 |
| 8 | B3 | 4684 | 11.7 | 1.55 | 3.39 | 2.60 | 54.65 | 58.89 |
| 8 | C1 | 7041 | 17.5 | 0.36 | 2.29 | 2.83 | 75.45 | 52.08 |
| 8 | C2 | 5412 | 13.5 | 1.40 | 3.09 | 4.01 | 74.23 | 58.42 |
| 8 | D | 3873 | 9.6 | 2.66 | 2.83 | 3.71 | 69.05 | 47.98 |

## Appendix B

**Table A2.** Table of the development of the classification of catchment groups produced by cluster analysis according to the degree of risk during the progressive formation of two to eight clusters.

| Number of Clusters | | Degree of Risk of Individual Parameters | | | | | Mean | Risk |
|---|---|---|---|---|---|---|---|---|
| | | SND | Tlag | Alfa | CN Mean | 6 h Rain | | |
| 2 | A | 1.68 | 1.07 | 0.92 | 1.09 | 0.95 | 1.14 | increased risk |
| | B | 0.65 | 0.92 | 1.14 | 0.88 | 1.07 | 0.93 | decreased risk |
| 3 | A1 | 0.90 | 0.67 | 0.68 | 1.05 | 0.95 | 0.85 | low risk |
| | A2 | 1.93 | 1.30 | 1.09 | 1.09 | 0.96 | 1.27 | high risk |
| | B | 0.65 | 0.99 | 1.20 | 0.86 | 1.08 | 0.96 | medium risk |
| 4 | A1 | 1.02 | 0.65 | 0.65 | 1.06 | 0.94 | 0.86 | decreased risk |
| | A2 | 2.31 | 1.29 | 1.09 | 1.10 | 0.93 | 1.35 | high risk |
| | B1 | 0.79 | 0.94 | 1.19 | 0.78 | 0.99 | 0.94 | decreased risk |
| | B2 | 0.61 | 1.11 | 1.07 | 1.03 | 1.16 | 0.99 | medium risk |
| 5 | A11 | 1.14 | 1.06 | 0.60 | 1.09 | 0.94 | 0.97 | medium risk |
| | A12 | 0.97 | 0.46 | 0.91 | 1.00 | 0.94 | 0.86 | decreased risk |
| | A2 | 2.31 | 1.30 | 1.15 | 1.10 | 0.94 | 1.36 | high risk |
| | B1 | 0.79 | 1.02 | 1.20 | 0.78 | 0.99 | 0.96 | medium risk |
| | B2 | 0.60 | 1.12 | 1.08 | 1.02 | 1.16 | 1.00 | medium risk |
| 6 | A11 | 1.56 | 1.05 | 0.59 | 1.08 | 0.95 | 1.05 | medium risk |
| | A12 | 1.03 | 0.45 | 0.91 | 1.00 | 0.94 | 0.87 | decreased risk |
| | A2 | 2.92 | 1.30 | 1.14 | 1.10 | 0.93 | 1.48 | high risk |
| | B1 | 0.89 | 1.00 | 1.22 | 0.76 | 1.01 | 0.98 | medium risk |
| | B2 | 0.79 | 1.11 | 1.10 | 1.02 | 1.19 | 1.04 | medium risk |
| | D | 0.41 | 1.15 | 0.99 | 1.02 | 0.95 | 0.91 | decreased risk |
| 7 | A11 | 1.48 | 1.02 | 0.57 | 1.09 | 0.95 | 1.02 | medium risk |
| | A12 | 1.00 | 0.45 | 0.90 | 1.01 | 0.96 | 0.86 | decreased risk |
| | A2 | 2.59 | 1.23 | 1.10 | 1.11 | 0.89 | 1.38 | high risk |
| | B1 | 1.10 | 1.04 | 1.10 | 0.79 | 0.93 | 0.99 | medium risk |
| | B3 | 0.63 | 0.95 | 1.37 | 0.81 | 1.19 | 0.99 | medium risk |
| | C | 1.24 | 1.27 | 1.10 | 1.09 | 1.11 | 1.16 | high risk |
| | D | 0.39 | 1.15 | 0.95 | 1.03 | 0.98 | 0.90 | decreased risk |
| 8 | A11 | 1.43 | 1.01 | 0.55 | 1.09 | 0.96 | 1.01 | medium risk |
| | A12 | 1.03 | 0.43 | 0.90 | 1.00 | 0.95 | 0.86 | decreased risk |
| | A2 | 2.11 | 1.13 | 0.98 | 1.11 | 0.86 | 1.24 | high risk |
| | B1 | 1.18 | 1.03 | 1.07 | 0.78 | 0.92 | 1.00 | medium risk |
| | B3 | 0.67 | 0.96 | 1.44 | 0.79 | 1.17 | 1.01 | medium risk |
| | C1 | 2.83 | 1.43 | 1.32 | 1.10 | 1.04 | 1.54 | high risk |
| | C2 | 0.74 | 1.06 | 0.93 | 1.08 | 1.16 | 0.99 | medium risk |
| | D | 0.39 | 1.16 | 1.01 | 1.00 | 0.95 | 0.90 | decreased risk |

**Appendix C**

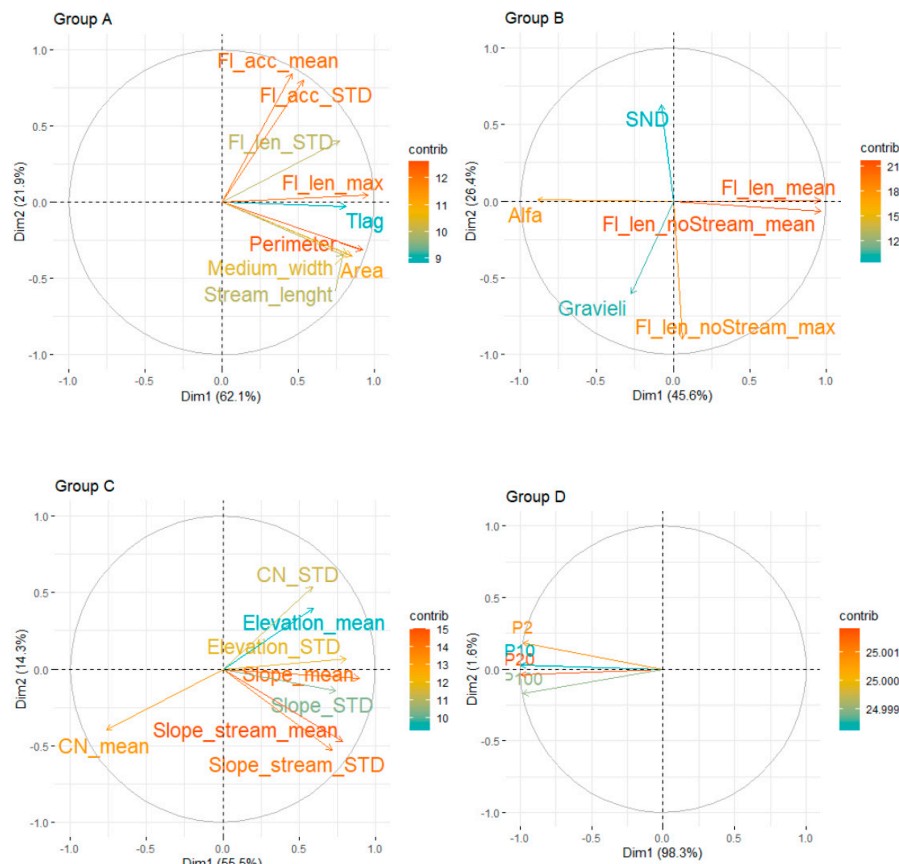

**Figure A1.** Biplots of the first two dimensions of the PCA analysis for Groups **A**–**D**.

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
