# Peer review of "Spatial Delimitation of Small Headwater Catchments and Their Classification in Terms of Runoff Risks"

_water, doi:10.3390/w13233458_

Round 1
Reviewer 1 Report
The authors revealed the spatial delimitation of small headwater catchments and their classifications with respect to runoff risks using cluster analysis. This work is interesting and within the scope of the journal. This study would be useful for the water resources and hydrology community. In general, this paper is well-written for most parts. However, a major revision is needed before it is published in the Water.
Specific comments:
- I think the author's summary of risk with respect to classifications of headwater catchments in the introduction is not sufficient. I suggest firstly provide some general information on this topic and then go to the case study of Czech Republic. This way would be more logical.
- I prefer to see a figure of study area and catchments’ locations as well as observations and DEM.
- Figure 1 and Table 3 presented the same message. I prefer to keep Table 3.
- Where are the sub-plots of Figure 5? Line 320-351.
- The legend is too small to read in Figure 6.
- The scale bar has too many fractions in Figure 6.
- The information reflected in Figure 7 is very fragmented and poor readable. Why not make a regionalization? Comprehensive regionalization has more practical reference value for hydrology modelling.
Author Response
Thank you for your comments to the draft article. Definitely contribute to a better understanding of the topic. Answer are here in text:
1. I think the author's summary of risk with respect to classifications of headwater catchments in the introduction is not sufficient. I suggest firstly provide some general information on this topic and then go to the case study of Czech Republic. This way would be more logical.
Thanks for the comments. They have been included in the article and perhaps this will improve the whole article.
2. I prefer to see a figure of study area and catchments’ locations as well as observations and DEM.
A picture of the site, including altitude hypsometry and location within Central Europe, has been added to the article
3. Figure 1 and Table 3 presented the same message. I prefer to keep Table 3.
Based on your recommendation, the discarded this duplication and reduced to a spreadsheet only.
4. Where are the sub-plots of Figure 5? Line 320-351.
The Sankey diagram shows the evolution of clusters. Value of centroids are in Appendix A. The picture link in text has wrong number. Nubers of figure and tab has been renumbered.
5. The legend is too small to read in Figure 6 (now is No. 8).
6. The scale bar has too many fractions in Figure 6.
The originally numbered Figure 6 has been modified. Legend are bigger and scalebar from the original Figure 6 has been removed. The scale is evident from Figures 1 and 10
7. The information reflected in Figure 7 is very fragmented and poor readable. Why not make a regionalization? Comprehensive regionalization has more practical reference value for hydrology modelling.
Based on the second reviewer's comments, a Clustering Validity Indices (CVI) was performed to determine the optimal number of clusters. A lower number of clusters is more appropriate. 5 cluster are optimal by CVI. Class D, which arises at six clusters, is specific and therefore the final number of clusters has been estimated to six in terms of risk. The regionalization would result in the loss of individual parameters of specific SHC and prevails the effort to preserve the information of each SHC and the geographical delimitation of the areas remains only in verbal form. In addition, regionalization itself would be an extension to another topic.
The data will be presented online at rain.fsv.cvut.
Reviewer 2 Report
The paper focuses on the classification of headwater catchments for rainfall-related hazard/risk purposes. Specifically, the first part of the paper focuses on the identification of relevant descriptive parameters such as slope, area and more complex indicators. The second part is devoted to the clustering of the catchments basing on a sub-set of those indicators, considered as independent variables. Overall, the topic is of interest. However, the paper currently lacks some “scientifical” content that could add further value for publication in a top-ranking journal as this one. I suggest major revisions addressing the following issues:
- In my opinion the most important issue relates to the concept that, potentially, every set of elements can be divided in clusters, but this does not mean that the resulting clusters are meaningful. The clustering operation itself is very simple to perform (K-means in this case) but the added value of clustering is not shown. For example, one of the key points is the choice of the number of clusters. Why does the Author imply that the clustering solution with K=8 clusters is better than, e.g., K=10? Similarly, is the Author sure that clustering with K>5 is worth the try? In my opinion additional evaluations should be presented concerning Clustering Validity Indices (CVI) such as the Davies-Bouldin Index or the Calinski-Harabasz Index. Such indices summarize clustering meaningfulness by comparing the within-cluster-distance against the between-cluster-distance. See Padulano et al. (2018) (ttps://doi.org/10.1007/s11269-018-2012-7) and Dimitriadu et al. (2002) (0033-3123/2002-3/1999-0709-A) for additional information and reference.
- My second concern relates to the selection of a sub-set of relevant indicators for the catchments from a larger set of parameters. Here the scope of this analysis is clear, and figure 2 is very nice. However, there is a little confusion in the name of the parameters, so that the discussion of this part is hard to follow. For example, it is stated that precipitation “P20” is chosen, but which is P20? I suppose it is “P_20yr_6h”. The same goes for all the other parameters. Here the Author could somehow improve the readability of this section. My suggestion is that indicators are first divided in groups with general labels (for example “catchment shape”, “CN”, “Precipitation” and so on) and then, for each group, specific indicators are provided (e.g. “CN_mean”, “CN_STD” and so on). Such labels could be extended to Table 2 and Figure 2 to improve readability. As for the “scientific” part, it is not clear how specific indicators are selected: for example, why T=20 yr for precipitation? My suggestion is to formalize the selection of the sub-set of independent parameters by means of some metrics or procedures: for example, the Principal Component Analysis could be used here.
Other minor issues:
- IDF curves. Some additional detail could be provided about how IDF curves were extracted from the radar precipitation info (e.g. time window, probability model). It is not clear whether this was done by the Author or IDF curves were already available from previous analysis.
- The paper would very much benefit of an extensive language check. There are several misspelling, wrong punctuation and repeated words/sentences (see for example caption of Figure 5).
- Figure 4. This figure is used to show that there is no prevalence of specific features according to the catchment size. In other words, the clustering according to size is not representative of the hydrological response. Although the figure is nice, it could be replaced by a similar figure representing the distribution of the same parameters in the different clusters of the final clustering solution (for example the one with the best CVI values). This would also improve the readability of the discussion part.
- The Reference section is quite short. Generally speaking, this is not a problem. However, the introduction section could be improved to better show the placement of this paper in the state-of-the-art, so to highlight that the tackled problem is worth concern and to further highlight the added value of the paper.
References
Dimitriadou, E., Dolničar, S., & Weingessel, A. (2002). An examination of indexes for determining the number of clusters in binary data sets. Psychometrika, 67(1), 137-159.
Padulano, R., & Del Giudice, G. (2018). A mixed strategy based on self-organizing map for water demand pattern profiling of large-size smart water grid data. Water Resources Management, 32(11), 3671-3685.
Author Response
1. In my opinion the most important issue relates to the concept that, potentially, every set of elements can be divided in clusters, but this does not mean that the resulting clusters are meaningful. The clustering operation itself is very simple to perform (K-means in this case) but the added value of clustering is not shown. For example, one of the key points is the choice of the number of clusters. Why does the Author imply that the clustering solution with K=8 clusters is better than, e.g., K=10? Similarly, is the Author sure that clustering with K>5 is worth the try? In my opinion additional evaluations should be presented concerning Clustering Validity Indices (CVI) such as the Davies-Bouldin Index or the Calinski-Harabasz Index. Such indices summarize clustering meaningfulness by comparing the within-cluster-distance against the between-cluster-distance. See Padulano et al. (2018) (ttps://doi.org/10.1007/s11269-018-2012-7) and Dimitriadu et al. (2002) (0033-3123/2002-3/1999-0709-A) for additional information and reference.
Clustering Validity Indices (CVI) was implemented in three variants. Only Davies-Bouldin made some obvious sense in determining the number of clusters. Although, from a statistical point of view, 5 seems to be an ideal number. Six clusters from hydrological possition are better some catchments switches between categories by one. Only 2% of catchemnts move from non-risk to risk.
2. My second concern relates to the selection of a sub-set of relevant indicators for the catchments from a larger set of parameters. Here the scope of this analysis is clear, and figure 2 is very nice. However, there is a little confusion in the name of the parameters, so that the discussion of this part is hard to follow. For example, it is stated that precipitation “P20” is chosen, but which is P20? I suppose it is “P_20yr_6h”. The same goes for all the other parameters. Here the Author could somehow improve the readability of this section. My suggestion is that indicators are first divided in groups with general labels (for example “catchment shape”, “CN”, “Precipitation” and so on) and then, for each group, specific indicators are provided (e.g. “CN_mean”, “CN_STD” and so on). Such labels could be extended to Table 2 and Figure 2 to improve readability. As for the “scientific” part, it is not clear how specific indicators are selected: for example, why T=20 yr for precipitation? My suggestion is to formalize the selection of the sub-set of independent parameters by means of some metrics or procedures: for example, the Principal Component Analysis could be used here.
Figure two was supplemented by groups of parameter groups analyzed below. PCA analysis is used for all data and for individual groups A-D. The results added to this paper. The contribution of individual parameters for "all data", especially in the case of the first two component, does not exceed 9% respective 13%.
Separatly for each group (A-D) first two component of the PCA analysis explain 80% of the variability for variable A, 70% for B and C. For group D (precipitation), the first component already 98 variability. The effect of the first two components are in biplots in new Appendix C.
The selected 5 parameters are not changed. A possible complaint can be directed primarily to the selection of two parameters for group C (SND and Alpha) instead of one parameter Flow_len_noStream_STD. Alone parameter is not significantly more explanatory and two parameters with a clearer link to hydrology have been preferred.
Other minor issues:
3. IDF curves. Some additional detail could be provided about how IDF curves were extracted from the radar precipitation info (e.g. time window, probability model). It is not clear whether this was done by the Author or IDF curves were already available from previous analysis.
Rainfall data are from previos work (now is two citations in introduction (Kašpar, Muller). I hope it is clearer after the text rewriting.
4. The paper would very much benefit of an extensive language check. There are several misspelling, wrong punctuation and repeated words/sentences (see for example caption of Figure 5).
I am not a native speaker, English was proof readed. I hope now is better.
5. Figure 4. This figure is used to show that there is no prevalence of specific features according to the catchment size. In other words, the clustering according to size is not representative of the hydrological response. Although the figure is nice, it could be replaced by a similar figure representing the distribution of the same parameters in the different clusters of the final clustering solution (for example the one with the best CVI values). This would also improve the readability of the discussion part.
t this point, I would consider it sufficient to show the values corresponding to the centroids for all gradually emerging clusters (tabel in appendix A). For your idea outside the article, I enclose boxplots for 5-7 clusters.
6. The Reference section is quite short. Generally speaking, this is not a problem. However, the introduction section could be improved to better show the placement of this paper in the state-of-the-art, so to highlight that the tackled problem is worth concern and to further highlight the added value of the paper.
Other relevant literature has been added, including your recommendation (CVI and PCA)

Round 2
Reviewer 1 Report
The auother has reponsed to my comments well. I think the manuscript could be accepted in publication but with a minor revision.
Specific comments:
- How many clusters exactly in Figure 10? Three or Six? The result is quite different with the last version.
- I still think the scale bar is too large in figures.
- Please add the elevation range in Figure 1.
- The names of rivers are too small to read in Figure 1.
Author Response
Answer:
- How many clusters exactly in Figure 10? Three or Six? The result is quite different with the last version.
Figure 10 does not show the number of clusters, but shows the degree of risk, for case of the 6 clusters - optimal count from CVI taking into account the hydrological potential response. Caption was rewrite for better comprehension
- I still think the scale bar is too large in figures.
- Please add the elevation range in Figure 1.
- The names of rivers are too small to read in Figure 1.
Scale bars was modified, elevation are added and river names are larger. I hope, now it is better.
Reviewer 2 Report
I am happy that the Author decided to include my suggestions in the revised version. In my opinion this increased the added value of the paper. However, there are still some minor issues that require attention.
Basically, when the Author removes the track change mode he will notice that a large number of misspelling/punctuation/interrupted sentence issues exist due to the changes and the introduction of new paragraphs. This occurs throughout the manuscript. In the new paragraphs, language issues often occur (missing articles, improper verb conjugation). Right now, the paper has a very messy appearance: I strongly recommend that a final revision and language proof reading is performed and uploaded for the Reviewers’ to see (track change mode off).
As for the scientifical part, I suggest that the novel concepts (PCA and CVI) are defined/explained, of course in short, but this would make the description of results clearer (especially the PCA part where new figures are also devoted to the description of related results). I also suggest that lines 342-358 are revised to improve readability (here there are also a lot of language issues which make reading hard) so that this part can be better related to the sentence in the Discussion part at lines 563-567. Also, the meaning of the following sentence at lines 567-569 is not clear: do you mean that you want to investigate this issue in future research? Finally, Figure in Appendix B is interesting but cannot be read, font size is too small.
Author Response
I am happy that the Author decided to include my suggestions in the revised version. In my opinion this increased the added value of the paper. However, there are still some minor issues that require attention.
Thank you for your comments, I tried to respond to them in the article and also here.
1.) Basically, when the Author removes the track change mode he will notice that a large number of misspelling/punctuation/interrupted sentence issues exist due to the changes and the introduction of new paragraphs. This occurs throughout the manuscript. In the new paragraphs, language issues often occur (missing articles, improper verb conjugation). Right now, the paper has a very messy appearance: I strongly recommend that a final revision and language proof reading is performed and uploaded for the Reviewers’ to see (track change mode off).
"Track change on" of the modified document are only in the .pdf, which is apparently generated automatically. I uploaded the version in .docx without revisions.
This version (v3) is after language proofreading. I hope the text is clearer now. Please look to the MSword version.
2.) As for the scientifical part, I suggest that the novel concepts (PCA and CVI) are defined/explained, of course in short, but this would make the description of results clearer (especially the PCA part where new figures are also devoted to the description of related results). I also suggest that lines 342-358 are revised to improve readability (here there are also a lot of language issues which make reading hard) so that this part can be better related to the sentence in the Discussion part at lines 563-567. Also, the meaning of the following sentence at lines 567-569 is not clear: do you mean that you want to investigate this issue in future research? Finally, Figure in Appendix B is interesting but cannot be read, font size is too small.
The concept of PCA and CVI and their use in the paper are at the end of the methodology section with reference to the relevant sources describing their own methods in more detail.
The marked passages have been rewritten, I hope for a better understanding. The font in the attachment image has been enlarged.